# Chalcone synthase (CHS) family members analysis from eggplant (*Solanum melongena* L.) in the flavonoid biosynthetic pathway and expression patterns in response to heat stress

**Xuexia Wu**[ID]°, **Shengmei Zhang**°, **Xiaohui Liu, Jing Shang, Aidong Zhang, Zongwen Zhu, Dingshi Zha**\*

Shanghai Key Laboratory of Protected Horticultural Technology, Horticultural Research Institute, Shanghai Academy of Agricultural Sciences, Shanghai, China

☉ These authors contributed equally to this work.
\* dingshizha1966@126.com

**Data Availability Statement:** All relevant data are within the manuscript and its Supporting Information files.

## Abstract

Enzymes of the chalcone synthase (CHS) family participate in the synthesis of multiple secondary metabolites in plants, fungi and bacteria. CHS showed a significant correlation with the accumulation patterns of anthocyanin. The peel color, which is primarily determined by the content of anthocyanin, is an economically important trait for eggplants that is affected by heat stress. A total of 7 *CHS* (*SmCHS1-7*) putative genes were identified in a genome-wide analysis of eggplants (*S. melongena* L.). The *SmCHS* genes were distributed on 7 scaffolds and were classified into 3 clusters. Phylogenetic relationship analysis showed that 73 *CHS* genes from 7 Solanaceae species were classified into 10 groups. *SmCHS5*, *SmCHS6* and *SmCHS7* were continuously down-regulated under 38˚C and 45˚C treatment, while *SmCHS4* was up-regulated under 38˚C but showed little change at 45˚C in peel. Expression profiles of key anthocyanin biosynthesis gene families showed that the PAL, 4CL and AN11 genes were primarily expressed in all five tissues. The CHI, F3H, F3'5'H, DFR, 3GT and bHLH1 genes were expressed in flower and peel. Under heat stress, the expression level of 52 key genes were reduced. In contrast, the expression patterns of eight key genes similar to *SmCHS4* were up-regulated at a treatment of 38˚C for 3 hour. Comparative analysis of putative CHS protein evolutionary relationships, *cis*-regulatory elements, and regulatory networks indicated that *SmCHS* gene family has a conserved gene structure and functional diversification. *SmCHS* showed two or more expression patterns, these results of this study may facilitate further research to understand the regulatory mechanism governing peel color in eggplants.

**Funding:** In addition, the authors have declared this work was supported by the Agricultural Committee Basic Project (Shanghai Agricultural word (2015) No 6-2-3), the National Key Technology R&D Program during the 13th Five-Year Plan Period (2017YFD0101904) and the China Agriculture Research System (Grant No. CARS-25). And the funding bodies supporting this work did not play a role in the design of the study and the collection, analysis, and interpretation of data or in the composition of the manuscript.

**Competing interests:** the authors have declared that no competing interests exist.

## Introduction

Eggplant (S. *melongena* L.) is one of the most important thermophilic vegetables produced in many tropical and temperate regions around the world. The optimum growth temperature for eggplant is between 22 and 30°C. After subjected to high temperature treatment, eggplants may exhibit to stagnation of growth, abortion of flower buds, and pollen viability rate and fruit set decrease, and the peel's color will turn light when the temperature is over 35°C. High temperature severely reduces the yield and affects the appearance quality of eggplant. However, the molecular mechanism of heat stress response in eggplants has not been thoroughly elucidated.

Anthocyanins are plant secondary metabolites and are among the most abundant natural pigments, that are responsible for the characteristic colors in flowers, fruits and vegetables plant tissues. The anthocyanin biosynthesis pathway has been studied in numerous plant species and most of the genes involved in this process have been identified. Moreover, anthocyanins play an important role in plant survival under stressful environmental conditions. High temperatures are known to reduce anthocyanin accumulation and have discoloration effects in many plant tissues, causing drastic effects in colored flowers [1, 2], and affecting the skin of such fruits as grape berries, apples and eggplant [3–7].

It is well known that CHS is the gatekeeper of the anthocyanin pathway [8]. Enzymes of chalcone synthase (CHS) are member of the plants-specific type III polyketide synthase (PKS) [9, 10], family and catalyze the first committed step of the branch of the phenylpropanoid pathway, which leads to the synthesis of flavonoids [11, 12]. Flavonoids are well known as a group of plant secondary metabolites that comprise several different classes of compounds, such as chalcones, flavones, flavonol isoflavones and anthocyanins. Flavonoids have a wide variety of biological functions in flower pigmentation, protection against UV radiation, pathogen defense, auxin transport and pollen fertility [13–15]. CHS also showed a significant correlation with synthesis of flavonoid compounds during heat stress defense. Heat stress responsive element in bread wheat (*Triticum aestivum* L.) has been found in the promoter of *Chs-D1* gene [16]. High-temperature stress had a large impact on the expression of *CHS7*, *CHS8* in both seeds and pods of Soybean [17]. The transcript levels of *CHS* decreased in apple peel and rose flower after heat treatment [1, 4]. In cork oak, *CHS* gene expression exhibited an increase under 45°C, but showed a decreased expression at 55°C [18]. The emergence of CHSV and CHSVII is important for the development of fungal heat stress tolerance and pathogenicity in pathogenic fungi. [19]. In addition, *CHS* (Sme2.5_00283.1_g00002.1) was up-regulated, and the other two *CHS* gene members were down-regulated under heat stress in peel of eggplant [7].

The product of the CHS reaction is a pivotal precursor for a large array of secondary metabolites derived from malonyl-CoA and p-coumaroyl-CoA. CHS exists as homodimeric iterative PKS (monomer size of 42–45 kDa) with two independent active sites that catalyze a series of decarboxylation, condensation, and cyclization reactions [10, 20]. Member of the CHS superfamily share high similarity in their amino acid sequence, which contains the structurally conserved catalytic center consisting of four residues, Cys-His-Asn-Phe, and most of the genes contain two exons and one intron [21]. However, the *CHS* gene family has not been characterized in eggplants to date.

In the current study, all *SmCHS* family members were identified in eggplant. A comprehensive analysis of members was performed, including gene structures, the biochemical characteristics of putative CHS protein, promoter *cis*-elements, phylogenetic relationships among members in other relative species, and their expression profiles in various organs/tissues

under high temperature stress. The findings of the present study may facilitate functional studies on eggplant *SmCHS* family genes.

## Materials and methods

### Plant materials and RNA extraction

The eggplant cultivar 'Tewangda' is a cold-tolerant cultivar with blackish purple skin. This cultivar grows vigorously and has good fruit setting. The fruit size was about 27.6 cm in length, 5.4 cm transverse diameter and a 209 g single fruit weight on average. 'Tewangda' fruits were grown at the same growth stage and were randomly selected. These plants were grown 144 days after sowing, and then placed inside incubators set at 27˚C (CK), 38˚C or 45˚C for 3 or 6 h (three plants per treatment). For each treatment, the tissue samples of root, stem, leaf, flower and peel were obtained and immediately frozen in liquid nitrogen and stored at -80˚C for RNA extraction and other analyses. All plant materials examined in this study were obtained from Shanghai Academy of Agricultural Sciences. Total RNA was extracted from each tissue sample using the mirVana miRNA Isolation Kit (Ambion) following the manufacturer's protocol. The extracted total RNA was stored at -80˚C. RNA integrity was evaluated using the Agilent 2100 Bioanalyzer (Agilent Technologies, Santa Clara, CA, USA).

### Identification of the CHS family members in the eggplant genome

The whole protein sequence of *Solanum melongena* L. (eggplant) were obtained from the Eggplant Genome DataBase (http://eggplant.kazusa.or.jp) [22], and those of *Solanum tuberosum* L. (potato, http://solanaceae.plantbiology.msu.edu/pgsc_download.shtml) [23], *Solanum lycopersicum* (tomato, https://solgenomics.net/organism/Solanum_lycopersicum/genome) [24], *Solanum penellii* (wild tomato, https://www.plabipd.de/project_spenn/start.ep) [25], *Capsicum annuum* L. (pepper, http://peppergenome.snu.ac.kr) [26], *Petunia axillaris* (https://solgenomics.net/organism/Petunia_axillaris/genome) [27], *Petunia inflate* (https://solgenomics.net/organism/Petunia_inflata/genome) [27], and *Nicotiana tabacum* (common tobacco, https://www.ncbi.nlm.nih.gov/nuccore/AYMY00000000) [28]. The profiles of CHS (PF00195 and PF02797) were downloaded from the Pfam protein family database (http://pfam.xfam.org/), and these profile sequences were used as queries to perform BLASTP searches against the protein sequence data of all the species mentioned above with a maximum E-value of $1 \times 10^{-3}$, respectively [29]. To further verify the exact copy number of CHS and remove redundant sequences, the Pfam database and Genome websites were also searched using "chalcone synthase" as keywords. All CHS sequences were submitted to EXPASy (https://web.expasy.org/protparam/) to calculate the number of amino acids, molecular weights and theoretical isoelectric points (pI).

### Structural characterization

The locations and intron numbers of CHS were acquired through the genome website. All of the acquired protein sequences were first aligned by ClustalX software with the default parameters [30]. An unrooted maximum-likelihood phylogenetic tree was constructed using MEGA6 software with a Bootstarp value of 1000 times [31]. The MEME program (Version 5.0.5, http://meme-suite.org/tools/meme) was used to identify the conserved motif of the CHS sequences with the following parameters: any number of repetitions, maximum of 10 misfits and optimum motif width of 6–200 amino acid residues. The WoLF PSORT program was used to predict the subcellular localization information of CHS proteins (https://www.genscript.com/wolf-psort.html) [32].

## Analysis of *cis*–acting elements in SmCHS

The upstream sequences (2 kb) of the *SmCHS* coding sequences in eggplant were retrieved from the genome sequence and then submitted to PlantCARE (http://bioinformatics.psb.ugent.be/webtools/plantcare/html/) to identify regulatory elements [33].

## Phylogenetic analysis of CHS genes

The full-length protein sequences of all eight species in Solanaceae were used for phylogenetic analysis. All of the protein sequences were first aligned by ClustalX software with the default parameters [30]. The phylogenetic tree was generated with MEGA6 software with a bootstrap test of 1000 times. The final tree was viewed and modified in Evolview software [34]. The CHS genes were classified into different groups according to the topology of the phylogenetic tree.

## Expression analysis of antyocyanin biosynthetic genes and construction of the mRNA regulatory network

The RNA-seq results were obtained by our lab [35]. Gene expression level was estimated from mean FPKM (fragments per kilobase of exon model per million reads mapped) values for each treatment, and showed the expression patterns in heatmap.Significant differentially expressed genes (fold change $\geq 2$ and *p*-value $\leq 0.05$) were used to calculate the Pearson correlation coefficient between *CHS* genes and other genes. The TBtools program was used to elucidate the Gene Ontology (GO) functional classification for the mRNAs with correlation coefficients greater than 0.9 [36]. The top 5 regulatory mRNAs annotated by GO enrichment for the genes associated with anthocyanin biosynthesis were collected to construct the regulatory network. The network was visualized using Cytoscape [37].

## qRT-PCR analysis

Expression level of anthocyanin biosynthesis genes, including phenylalanine ammonia lyase (PAL), cinnamate 4-hydroxylase (C4H), 4-coumarateCoA ligase (4CL), CHS, chalcone isomerase (CHI), flavanone 3-hydroxylase (F3H), flavonoid 3′-hydroxylase (F3′H), flavonoid 3′5′-hydroxylase (F3′5′H), dihydroflavonol4-reductase (DFR), anthocyanidin synthase (ANS), anthocyanidin 3-O-glucosyltransferase (3GT), Anthocyanins11 (AN11), and most transcription factors, such as myeloblastosis (MYB), basic helix-loop-helix (bHLH) and metadata authority description schema(MADS1), were analyzed. First-strand cDNA was synthesized from 1 μg total RNA from 5 tissues (root, stem, leaf, flower and peel) using a Prime Script RT Reagent Kit (Takara, Dalian, China). The qRT-PCR reactions were performed in 96-well plates using the ABI 7500 fast Real-Time PCR system (Applied Biosystems, USA) with the Quanti-Fast SYBR Green PCR Kit (Qiagen, Duesseldorf, Germany). The qRT-PCR parameters were as follows: 95˚C for 5 min, then 45 cycles of 95˚C for 10 s, 60˚C for 10 s, and 72˚C for 10 s. The relative mRNA expression levels were calculated using the $2^{-\Delta\Delta CT}$ method [38]. PGK (JX154676) was used as an internal control to normalize the data. For each sample, three biological repeats were performed, the relative expression levels were calculated using the standard curve and normalized by the control's expression, the results were display by heatmap. The primer sequences are listed in S3 Table.

## Results

### Identification of *CHS* genes and sequence analysis in Solanaceae species

A total of 7 *CHS* (*SmCHS1-7*) genes in eggplant were identified after being verified by protein sequence analysis and BlAST search using the eggplant genome annotation database (S1A Table). The length of SmCHS protein ranged from 327 to 396 amino acids (Table 1, S2 Table). The PKS type III active sites of the enzymes and Phe215 connected with CoA binding are conserved among all SmCHS (S1 Fig). In addition, 66 *CHS* genes were characterized from 7 other Solanaceae species. The subfamily numbers of *CHS* genes ranged from 6 (*Solanum penellii*) to 13 (*Petunia axillaris*) (Table 1, S1B–S1H Table). The length for the other 7 Solanaceae species (*Solanum tuberosum* L., *Solanum lycopersicum*, *Solanum penellii*, *Capsicum annuum* L., *Petunia axillaris*, *Petunia inflate* and *Nicotiana tabacum*) proteins ranged from 156 to 431 amino acids (S1A–S1G Table). The average number of amino acids was calculated, and all the amino acids of each species are arranged in same order to form a data set. The correlation coefficients among the above data set were all greater than 0.99. This finding suggests that *CHS* genes are conserved in Solanaceae species.

### Structure and conserved motif analysis of *SmCHS*

The 7 *SmCHS* genes were distributed on 7 scaffolds. To better understand the evolution of *SmCHS* genes, an unrooted maximum-likelihood tree was constructed based on the 7 SmCHS protein sequences, and the *SmCHS* were classified into 3 clusters (i, ii and iii) (Fig 1). Among the *SmCHS* genes, only one *SmCHS7* had three exons, and the others had two exons (Fig 1) based on information available from the genome annotation. These results suggest the potential diversity of the biological functions of the *SmCHS* genes in eggplants, previous studies also have similar conclusion [39, 40].

The phylogenetic tree was on the left of the figure and showed that the *SmCHS* genes were classified into three clusters (i, ii and iii). The exon/intron organization of *SmCHS* is shown on the right of the figure. For *SmCHS* gene organization, yellow boxes represent exons, black lines represent introns, and green boxes indicate upstream/downstream regions. The lengths of the exons and introns are drawn to scale.

To understand the functional diversification of *SmCHS*, the conserved motifs of these 7 protein sequences were identified by the MEME program, and 10 conserved motifs were detected in eggplant (Fig 2, Table 2). The Chal_sti_synt_C domain and Chal_sti_synt_N domain were included in motifs 1 and motifs 2, respectively. For all 7 eggplant SmCHS proteins, Motif 1 and motif 2 exist in all of them, motif 3 is only absent in *SmCHS5*, and motif 4 and motif 5 are only absent in *SmCHS1*. The N-terminal domain (PF00195) of the CHS protein contained motif 1 and the combination of motifs 3, 4, 6, 7 and 9. The C-terminal domain (PF02797) of the CHS protein contained motif 2 and the combination of motifs 5, 8 and 10. Therefore, the motif configuration of the SmCHS reflects the conservation and diversity of the CHS family. To further investigate the subcellular localization information of SmCHS proteins, the WoLF PSORT program was used to predict the localization of SmCHS protein [31]. SmCHS7 was predicted to localize in the nucleus, whereas SmCHS4 and SmCHS6 were predicted to localize in the chloroplast. The others SmCHS proteins were predicted to localize in the cytoplasm. The different compositions of the domains and subcellular localization may indicate functional diversity.

### Stress-related *cis*-elements in *SmCHS* promoters

To further study the potential regulatory mechanisms of *SmCHS* during abiotic stress responses, the 2 kb upstream sequences from the translation start sites of *SmCHS* were used to

**Table 1. Features of *SmCHS* genes identified in eggplant.**

| Gene Name | Gene ID | Number of amino acids |
|---|---|---|
| *SmCHS1* | Sme2.5_01077.1_g00016.1 | 333 |
| *SmCHS2* | Sme2.5_02154.1_g00001.1 | 389 |
| *SmCHS3* | Sme2.5_13923.1_g00001.1 | 389 |
| *SmCHS4* | Sme2.5_00283.1_g00002.1 | 392 |
| *SmCHS5* | Sme2.5_01039.1_g00002.1 | 327 |
| *SmCHS6* | Sme2.5_00346.1_g00019.1 | 396 |
| *SmCHS7* | Sme2.5_05261.1_g00004.1 | 383 |

identify the *cis*-elements (Fig 3A). The results showed that all *SmCHS* had common upstream promoter elements, including TATA-box and CAAT-box, which occurred more than 100 times; therefore, these sequences were presumed to be the promoter sequences (Fig 3B). The elicitor response element (ERE) and myeloblastosis binding *cis*-elements (MYB) occurred more than 10 times in the *SmCHS* upstream sequences. Research has shown that an increase in CHS activity causes a high accumulation of flavonoids that inhibits polar auxin transport [8, 41, 42]. Two *cis*-acting elements (ABRE, involved in abscisic acid responsiveness; AuxRR, involved in auxin responsiveness) were found in the upstream regions. MYB and myelocyto-matosis (MYC) binding sites have also been identified, which may greatly influence plant stress tolerance. Cluster analysis of *cis*-element number showed that 7 *SmCHS* genes were divided into 3 groups (I, II, III), and *SmCHS1*, *SmCHS2* and *SmCHS3* had similar regulatory pattern (Fig 3A). Five *cis*-elements (CARE, GCN4-motif, GT1-motif, MRE and TCT-motif) exist only in group I, GARE-motif only exist in group III. STRE exist in group II and III. These results showed that *SmCHS* is activated by a wide range of environmental and developmental stimuli, and there are many complex means of regulating *SmCHS* activity in eggplants.

## Phylogenetic analysis of *CHS* genes in Solanaceae

To analyze the evolutionary relationships of *CHS* genes in Solanaceae, an unrooted phyloge-netic tree was constructed using full-length amino acid sequences. All 73 *CHS* genes were clas-sified into 10 groups (Fig 4, Table 3), and the number of CHS gene groups ranged from two to

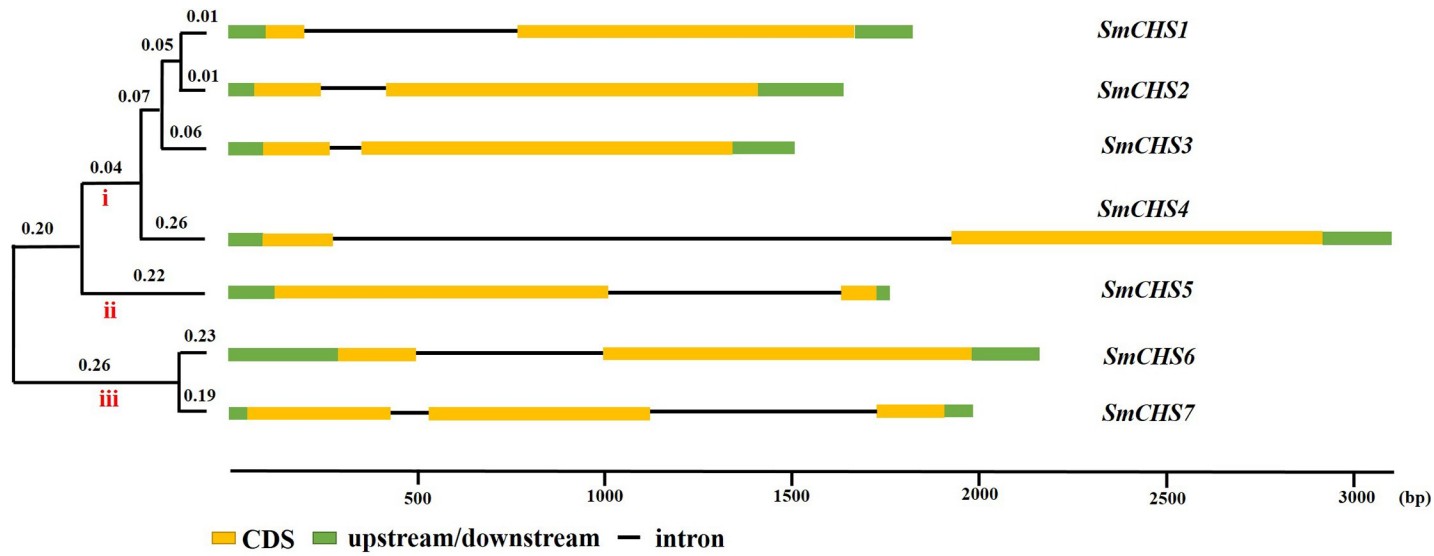

**Fig 1. Phylogenetic relationship and gene structure analysis of *SmCHS* genes.**

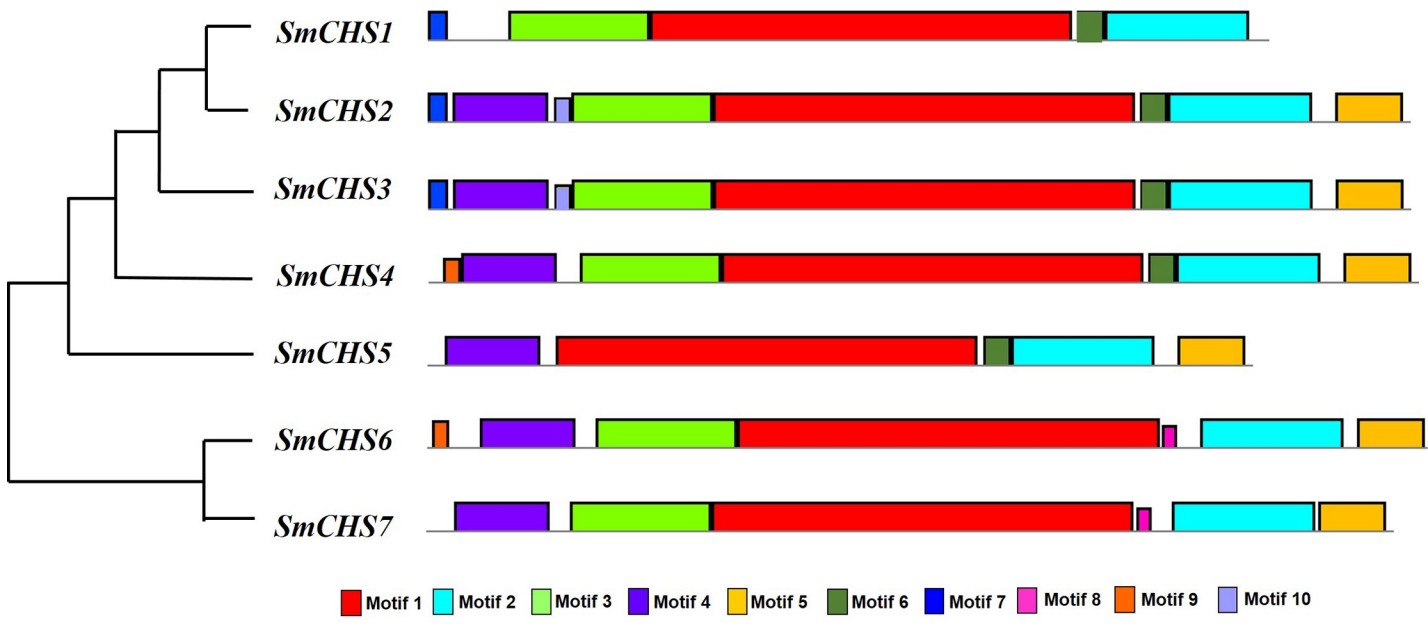

**Fig 2. Motifs conserved across all CHS proteins in eggplant.** Ten conserved motifs are indicated in differently colored boxes.

eleven. The 7 SmCHS were categorized into 6 groups (groups I, II, VII, VIII, IX and X), and group II contained *SmCHS1* and *SmCHS2*. Groups I, II, IX and X exist in all eight species, and groups III, IV and V were absent in *Solanum melongena* L., *Solanum penellii*, *Solanum lycopersicum* and *Solanum tuberosum* L.. The group VI is absent in *Capsicum annuum* L., *Nicotiana tabacum*, *Petunia inflate* and *Petunia axillaries* (Table 3). The VIII, IX and X groups are distinguished from other groups mainly depends on the position 1–164 amino acids, GroupsI, II and III are relatively conservative at the position 260–360 amino acids, in which the other groups are very diverse(S1 Fig). These results suggested that the CHS were conserved, but small variations existed among the eight species in Solanaceae and showed that SmCHS1, SmCHS2 and SmCHS3 were more conserved than SmCHS4 according to the phylogenetic tree.

## Expression profile of key anthocyanin biosynthesis genes in eggplants under heat stress

Using the RNA-seq data of eggplant peel, a heatmap of 96 key anthocyanin biosynthesis genes (PAL, C4H, 4CL, CHS, CHI, F3H/F3'H, F3'5'H, DFR, ANS, 3GT, MYB1, MYB2, bHLH1, AN11, MADS1) was established under heat stress (Fig 5). The expression of anthocyanidin synthase (ANS) and MYB2 was not identified during this sampling period (the undetected genes were also color-coded for 0 in Fig 5). For seven *SmCHS* genes, expression of three genes(*SmCHS5*, *SmCHS6*, and *SmCHS7*) were not identified, and the other four *SmCHS* genes were divided into two groups according to their expression patterns. Three of those four *SmCHS* genes (*SmCHS1*, *SmCHS2*, and *SmCHS3*) were continuously down-regulated under 38°C and 45°C treatment compared with the CK. However, *SmCHS4* was up-regulated under 38°C, but showed little change under 45°C in peel. These phenomena have also been observed in some other key gene families associated with anthocyanin biosynthesis. According to the RNA-seq results of 96 anthocyanin biosynthesis key genes in eggplant peel, *SmCHS4* showed the highest expression level at the 38°C-3h along with eight other genes (*Sme2.5_03336.1_g00008.1_PAL, Sme2.5_00041.1_g0 0017.1_4CL, Sme2.5_00283.1_g00002.1_smCHS4, Sme2.5_00298.1_g00002.1_F3H, Sme2.5_020*

**Table 2. List of the putative motifs of CHS proteins.**

| Motif | Length (amino acids) | Best possible match |
|---|---|---|
| Motif 1 | 167 | IKEWGQPKSKITHLVFCTTSGVDMPGADYQLTKLLGLRPSVKRFMMYQQGCFAGGTVLRL AKDLAENNKGARVLVVCSEITAVGFRGPSETHPDSLVGQA |
| Motif 2 | 57 | DWNSJFWIAHPGGPAILDQVELKLGLKPEKLRATRQVLSDYGNMSSACVLFILDEMR |
| Motif 3 | 56 | RLCDKSMIKKRYMHLTEEILKENPNLCEYMAPSLDARQDIVVVEVPKLGKEAAQKA |
| Motif 4 | 38 | QRAEGPATILAIGTATPSNCVDQSTYPDYYFRITNSEH |
| Motif 5 | 27 | TTGEGLDWGVLLGFGPGLTIETIVLHS |
| Motif 6 | 11 | LIEAFEPLGIS |
| Motif 7 | 8 | MVTVEEVR |
| Motif 8 | 6 | FCEKLI |
| Motif 9 | 7 | QNIGKVN |
| Motif 10 | 7 | ELKEKFK |

66.1_g00012.1_F3H, Sme2.5_04260.1_g00001.1_F3H, Sme2.5_15970.1_g00001.1_F3H, Sme2.5_00670.1_g00012.1_DFR, Sme2.5_00747.1_g00013.1_AN11) (Fig 6). In particular, Sme2.5_03336.1_g00008.1_PAL expression level under 38˚C doubled but was down-regulated at 45˚C compared with CK; Sme2.5_00670.1_g00012.1_DFR, Sme2.5_00747.1_g00013.1_AN11 expression level increased 3–4 fold and 7–10 fold under 38˚C, respectively. These results suggest that these genes may have tissue-specific or functionally differentiation.

## mRNA regulatory network associated with anthocyanin biosynthesis in eggplant

Pearson correlation coefficient was calculated between all the mRNAs and anthocyanin bio-synthesis related genes, the mRNAs with correlation coefficient greater than 0.9 are considered to be genes that are co-expressed with the anthocyanin biosynthesis genes. A total of 4928 mRNA correlation coefficients were more than 0.9, and all of these mRNAs were functionally categorized in the GO database. The top 20 GO enrichment results of biological processes are shown in Table 4. The genes involved in the regulation of biological processes (GO:0050789), regulation of cellular metabolic processes (GO:0031323) and regulation of gene expression (GO:0010468) were collected and filtered to construct a regulatory network. In totally, 67 anthocyanin biosynthesis key genes and 146 regulatory mRNAs were included in this regula-tory network (S2 Fig). These GO enrichment results suggest that the anthocyanin biosynthesis pathway may be regulated by a wide range of environmental and developmental stimuli.

## Expression pattern of anthocyanin biosynthesis key genes in different tissues under heat stress

Using the qRT-PCR data, a heatmap of 20 key anthocyanin biosynthesis genes was established in different tissues under heat stress (Fig 7). The qRT-PCR results showed a high consistency with the RNA-seq data, which suggested that the RNA-seq data were credible. Most of the CHS genes were expressed in peel and were expressed at low levels in other tissues. The PAL, 4CL and AN11 genes were mainly expressed in all five tissues. The CHI, F3H, F3′5′H, DFR, 3GT and bHLH1 genes were expressed in flower and peel. MADS1 was expressed in stems, leaves, flowers and peels. Under heat stress, cluster i (cluster show in Fig 2) was continuously downregulation, cluster ii was up-regulated 4 times under 38˚C compared with CK in peel, and cluster iii was not detected in most eggplant tissues.

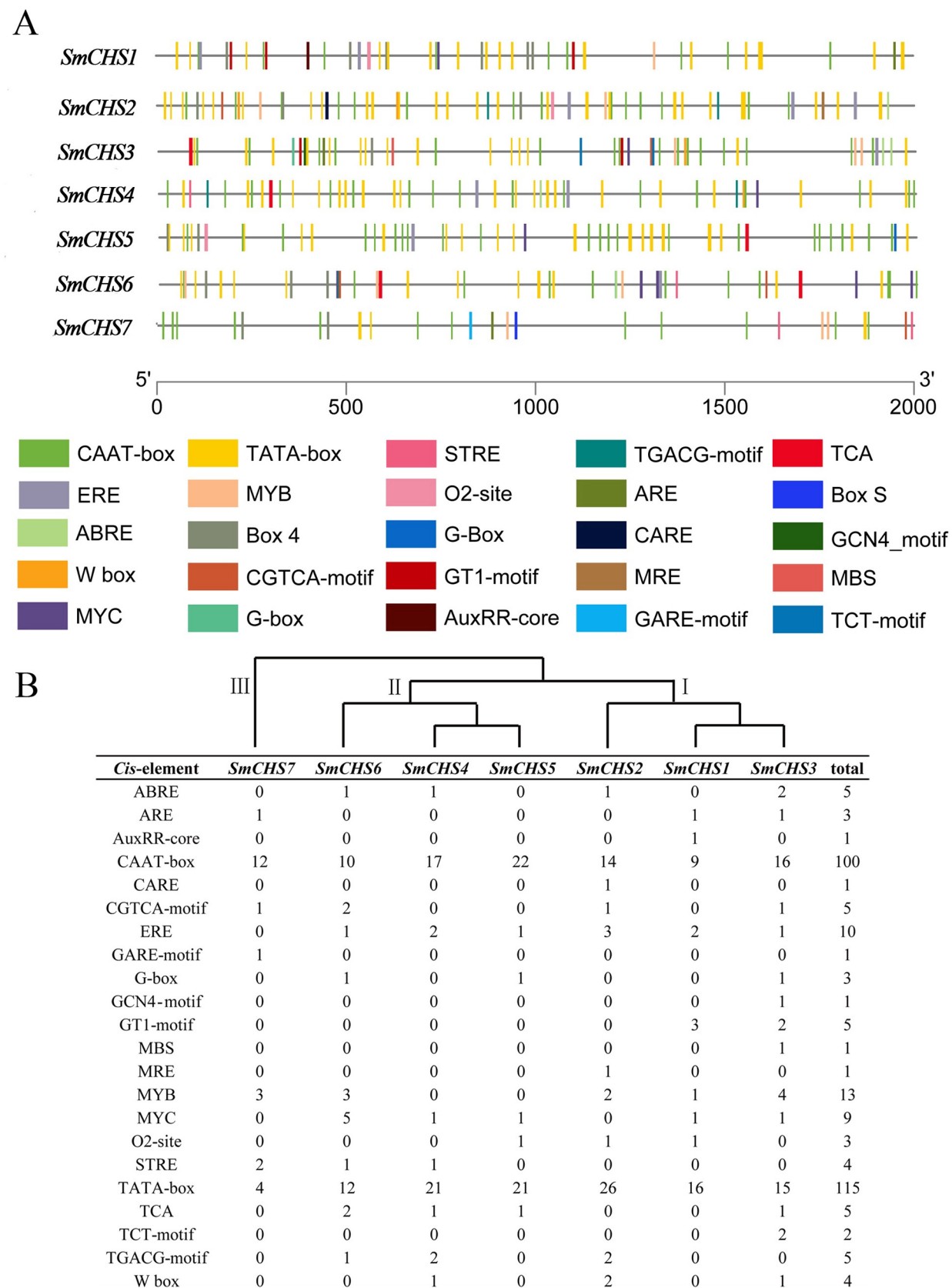

**Fig 3. *Cis*-elements in *CHS* family gene promoters.** (A) Frequency of cis-element occurrence in upstream sequences. (B) Predicted *cis*-elements in *CHS* gene promoters. The scale bar indicates the length of promoters.

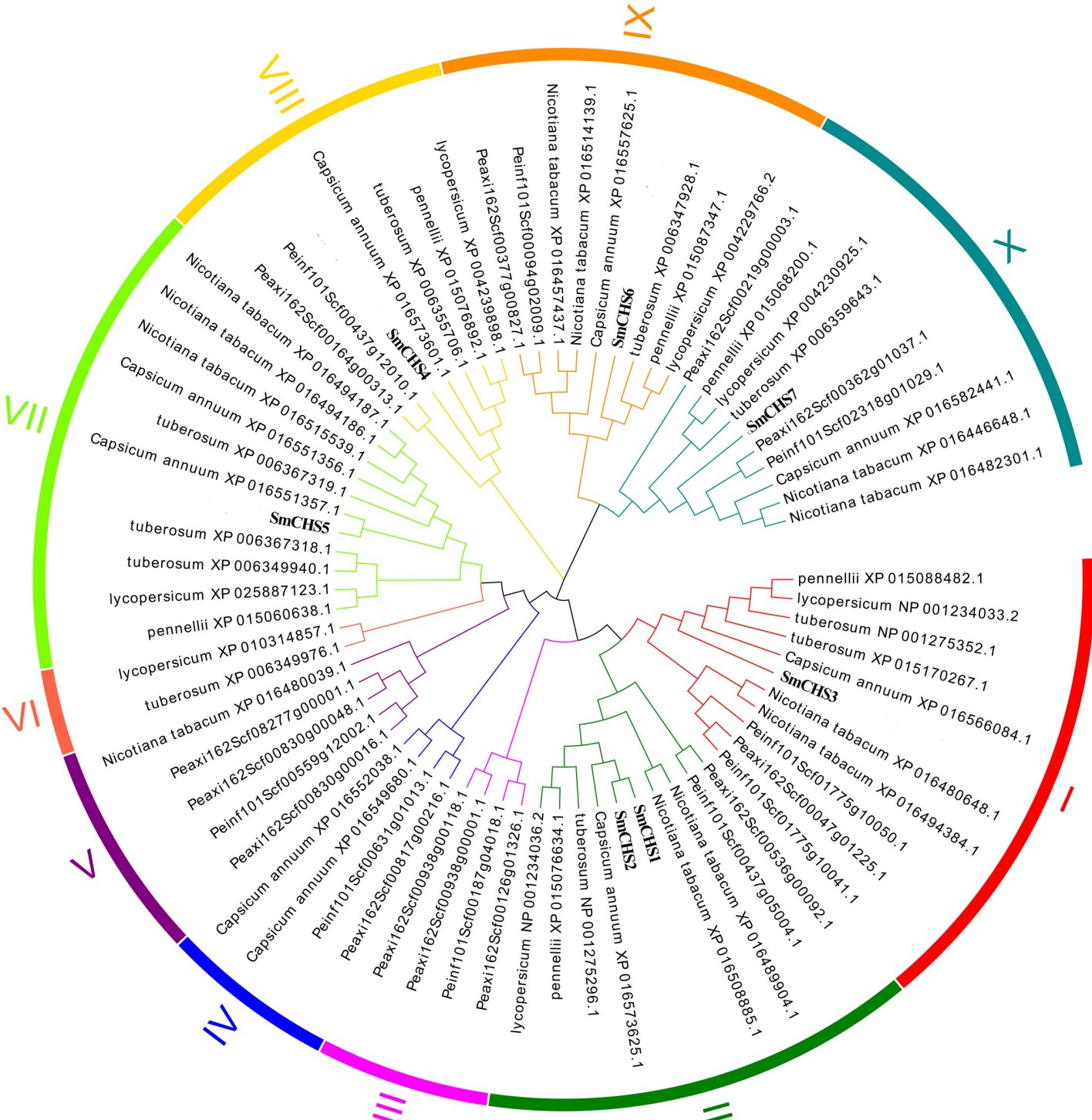

**Fig 4. Phylogenetic tree of *CHS* genes in Solanaceae species.** The colored region is associated with 10 groups of proteins (Group I to X).

**Table 3. Distribution of *CHS* genes in the phylogenetic tree.**

| Plant Pecies | Number | Phylogenetic Group | | | | | | | | | |
|---|---|---|---|---|---|---|---|---|---|---|---|
| | | I | II | III | IV | V | VI | VII | VIII | IX | X |
| *Solanum melongena* L. | 7 | 1 | 2 | 0 | 0 | 0 | 0 | 1 | 1 | 1 | 1 |
| *Solanum penellii* | 6 | 1 | 1 | 0 | 0 | 0 | 0 | 1 | 1 | 1 | 1 |
| *Solanum lycopersicum* | 7 | 1 | 1 | 0 | 0 | 0 | 1 | 1 | 1 | 1 | 1 |
| *Solanum tuberosum* L. | 10 | 2 | 1 | 0 | 0 | 0 | 1 | 3 | 1 | 1 | 1 |
| *Capsicum annuum* L. | 9 | 1 | 1 | 0 | 2 | 0 | 0 | 2 | 1 | 1 | 1 |
| *Nicotiana tabacum* | 12 | 2 | 2 | 0 | 0 | 1 | 0 | 3 | 0 | 2 | 2 |
| *Petunia inflate* | 9 | 2 | 1 | 1 | 1 | 1 | 0 | 0 | 1 | 1 | 1 |
| *Petunia axillaris* | 13 | 1 | 1 | 3 | 1 | 3 | 0 | 0 | 1 | 1 | 2 |

## Discussion

It is well-known that the *CHS* gene family plays a significant role in the growth and development of plants. In many species, multigene families of *CHS* have been identified. For example, six *CHS* genes have been described in turnip [43]. In maize, 14 complete *CHS* genes have been identified [39]. A total of 27 *CHS* genes were found in rice [40]. These studies showed that *CHS* members were divided into two or more subclasses according to phylogenetic analysis. Generally, genes grouped into the same subclasses shared similar evolutionary features, and obtained the same expression pattern. In our study, the identified sequences showed a high level of coding sequence similarity (above 90%). The *SmCHS* were classified into three clusters based on the results of the maximum-likelihood tree. At 35˚C, previous studies showed that *SmCHS1* and *SmCHS3* (Sme2.5_01077.1_g00016.1, Sme2.5_13923.1_g00001.1) were down-regulated in peels of eggplant [7], which is in keeping with our results, other two clusters *CHS* genes show different expression patterns. These results suggest the functional diversification of *SmCHS*.

Flavonoids have numerous functions and contribute to pigments, signaling molecules, and protectants against biotic and abiotic stresses. The flavonoid biosynthetic pathway is one of the most intensively investigated pathways for applied biological and genetic processes, as well as for understanding gene regulation, characterizing transposable elements and producing of agronomically stress-tolerant plants and natural dietary antioxidants. Biosynthesis of anthocyanins responds to environmental stress factors, such as light, nutrient depletion, and temperature change. The peel color determined by the content of anthocyanin is a majority economically important trait for eggplant, and this color is modulated by the genes in the flavonoid biosynthesis pathway. Compared with other tissues, *SmMYB1* and all anthocyanin biosynthetic key genes (*SmCHS*, *SmCHI*, *SmF3H*, *SmDFR*) except *SmPAL* were dramatically up-regulated in the fruit skin of the purple cultivar [44]. The full length cDNA of *SmCHS*, *SmCHI*, *SmF3'5'H*, and *SmDFR* were isolated from eggplants by Jiang (2016) [45]. These genes have the highest expression levels in peels except for *SmF3H*, which was detected in stems [45]. The expression profiles of these key gene families under heat stress were investigated in our study. 'These anthocyanin biosynthesis key genes (PAL, 4CL, AN11, CHI, F3H, F3'5'H, DFR, 3GT and bHLH1) show tissue specific expression, suggesting that these genes respond at the late stage of the anthocyanin pathway and directly regulate the color of fruit skin and flower.

Heat stress reduced the anthocyanin content and the enzyme activities of CHS, DFR, ANS and 3GT/UFGT in eggplant peel and strengthened the activity of PAL [35, 46]. When the temperature exceeds 35˚C, the eggplant will be dehydrated and shrink, and the peel's color will lighten. CHS is a key enzyme of the flavonoid biosynthesis pathway. Most of the genes

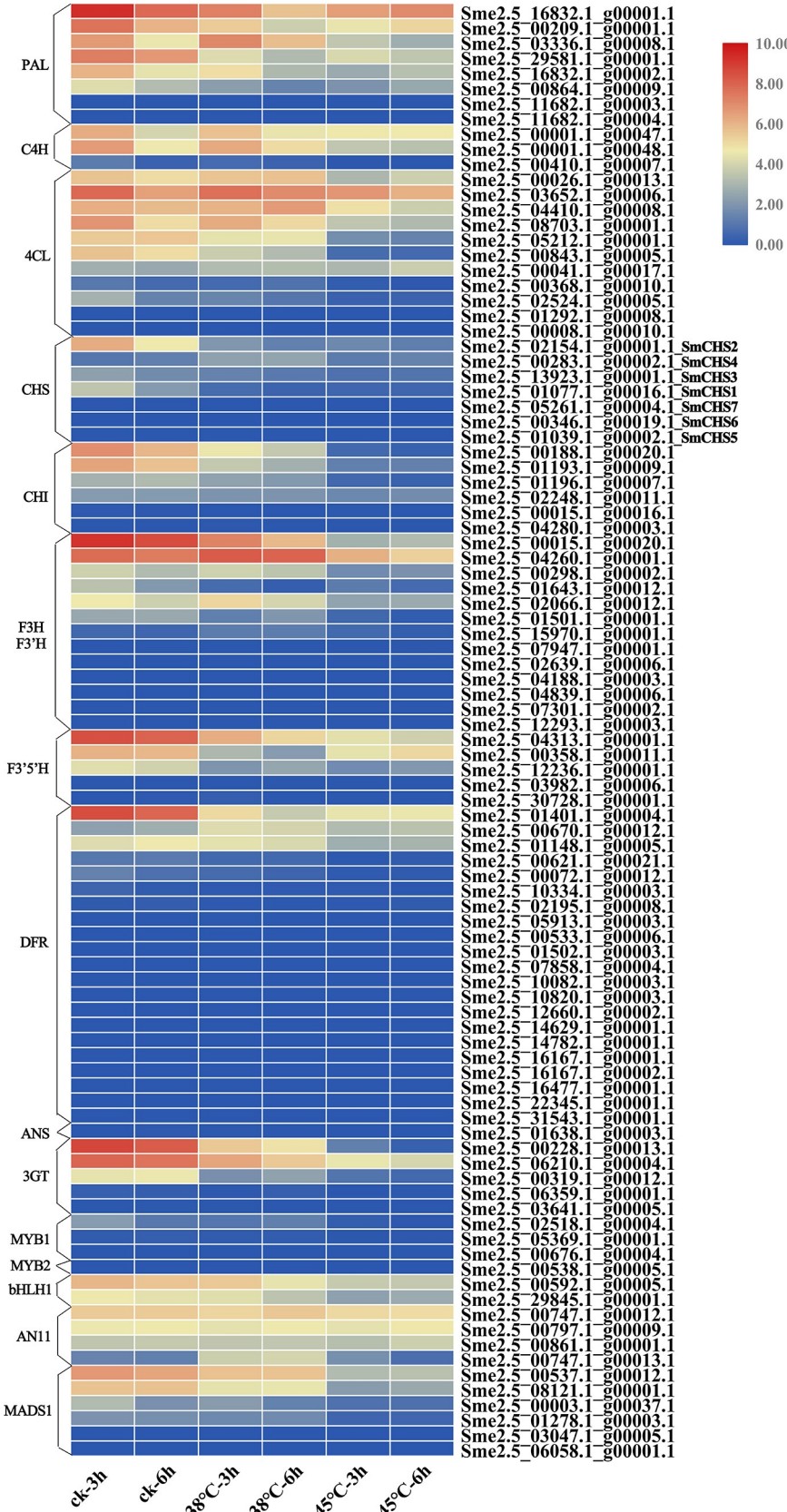

**Fig 5. Heatmap of 96 key anthocyanin biosynthesis genes expression level in eggplants peel under heat stress.** The color box from blue to red indicate an increased expression level.

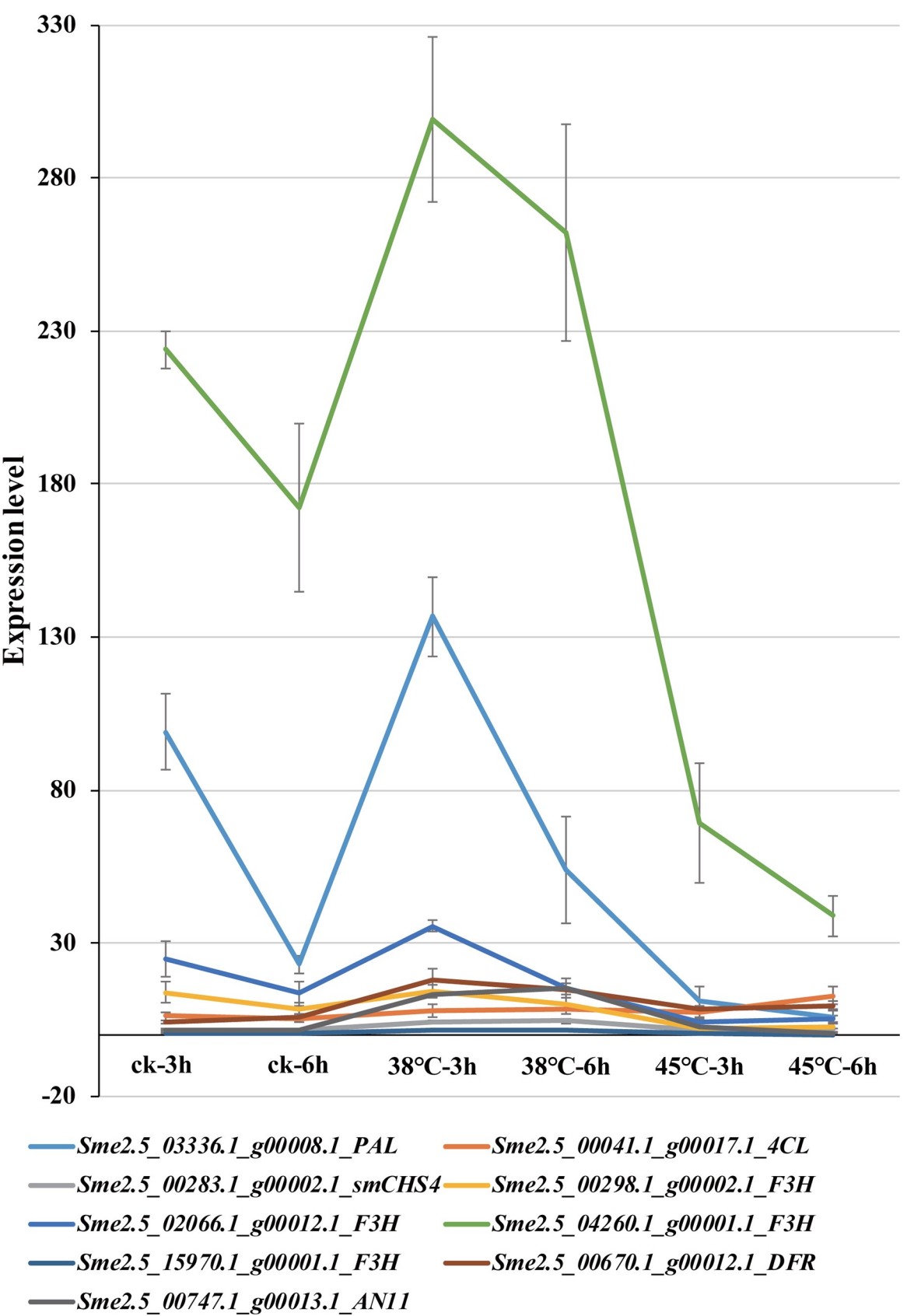

**Fig 6. Expression profiles of *SmCHS4* and eight anthocyanin biosynthesis genes in response to heat stress.** These genes have the highest expression level at 38˚C-3h in eggplant peel. The error bars represent the standard error of the means of three biological replicates.

associated with flavonoid biosynthesis were down-regulated under heat stress. In this study, the genes of the flavonoid biosynthesis pathway showed tissue-specificity, and genes expressed in different phases and tended to change over time (Fig 7). Under heat stress, *SmCHS4* and some anthocyanin biosynthesis related genes show different expression profiles at 38˚C-3h (Fig 6), suggest that these co-up-regulated genes contribute to protect the eggplant at beginning of heat stress defense. In addition, 52 gene expression levels were reduced under heat stress, which was similar to Lv's (2019) results [7], while 35 gene expression levels were not identified. These results suggest that some key anthocyanin biosynthesis genes help to protect the eggplant from damage to heat stress. Moreover, these gene families exhibited two or more expression patterns and performed multiple genetic functions to regulate anthocyanin content. Combined with regulatory networks, it is possible to further understand the regulatory mechanism of peel color in eggplants.

## Conclusions

In this study, a genome-wide analysis of the *SmCHS* gene family in eggplants was performed. The CHS protein biochemical characteristics, phylogenetic relationships, gene structures, *cis*-regulatory elements, regulatory network and functional predictions of the *smCHS* gene family members were examined. The *SmCHS* gene family has conserved gene structure and functional diversification. CHS plays important roles in the anthocyanin biosynthesis pathway, exhibits two or more expression patterns and executes multiple functions to regulate anthocyanin content in eggplant peels under heat stress. The result of this study may contribute to the

**Table 4. Top 20 GO enrichment results of biological processes.**

| GO term | GO ID | P value |
| --- | --- | --- |
| cellular biosynthetic process | GO:0044249 | 0 |
| cellular nitrogen compound biosynthetic process | GO:0044271 | 0 |
| cellular response to chemical stimulus | GO:0070887 | 0 |
| cellular response to stress | GO:0033554 | 0 |
| regulation of biological process | GO:0050789 | 0 |
| regulation of cellular macromolecule biosynthetic process | GO:2000112 | 0 |
| developmental process | GO:0032502 | 0 |
| regulation of RNA biosynthetic process | GO:2001141 | 0 |
| regulation of cellular metabolic process | GO:0031323 | 0 |
| cellular component organization | GO:0016043 | 0 |
| response to organic substance | GO:0010033 | 1.11E-16 |
| protein metabolic process | GO:0019538 | 1.11E-16 |
| regulation of nitrogen compound metabolic process | GO:0051171 | 1.11E-16 |
| regulation of gene expression | GO:0010468 | 1.11E-16 |
| response to stimulus | GO:0050896 | 1.11E-16 |
| regulation of nucleobase-containing compound metabolic process | GO:0019219 | 1.11E-16 |
| response to chemical | GO:0042221 | 1.11E-16 |
| cell communication | GO:0007154 | 1.11E-16 |
| response to stress | GO:0006950 | 1.11E-16 |
| oxoacid metabolic process | GO:0043436 | 2.22E-16 |

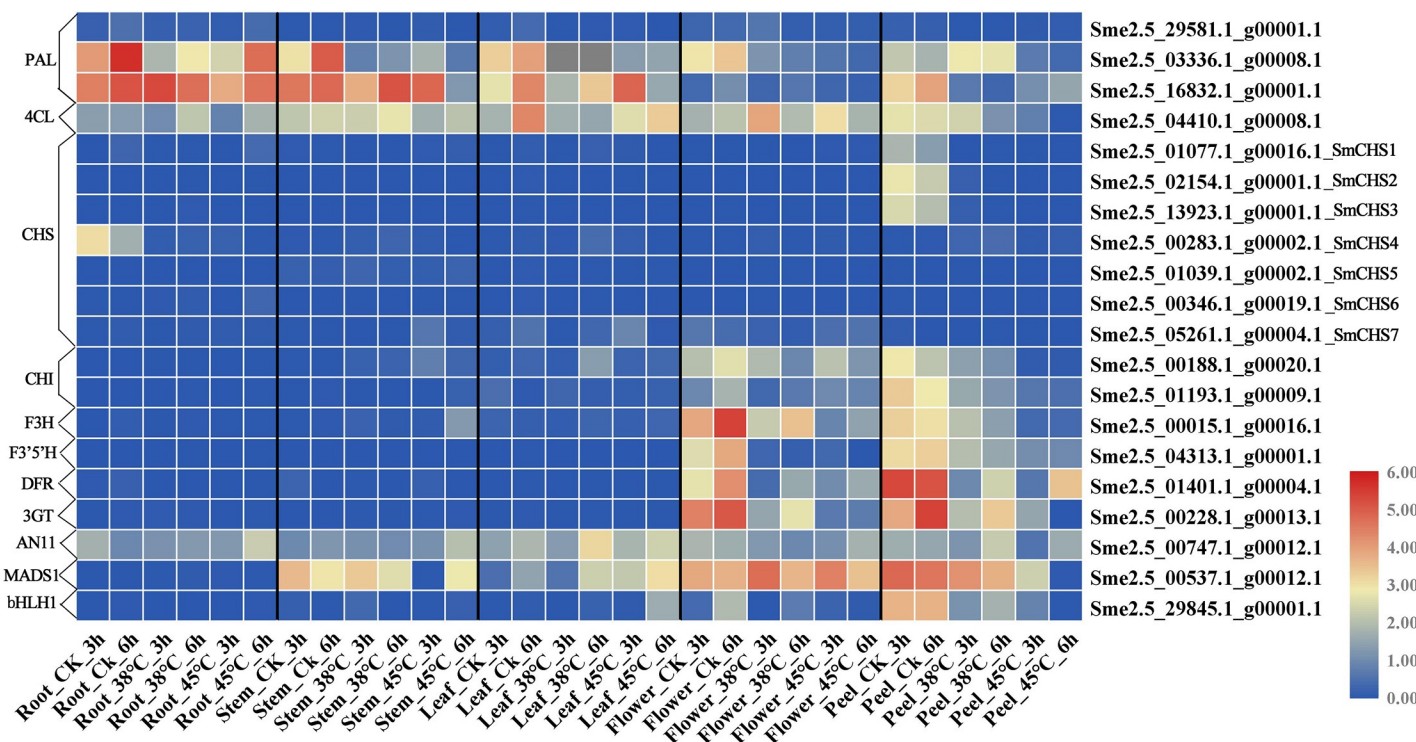

**Fig 7. Expression profiles of 20 key anthocyanin biosynthesis genes in different tissues.**

production of heat-resistant eggplant for further research on the functions, regulation and evolution of the CHS family.

## Supporting information

**S1 Table. CHS protein sequences of Solanum species.**
(XLSX)

**S2 Table. Features of CHS genes identified in Nicotiana tabacum.**
(XLSX)

**S3 Table. Primers used for real time PCR analysis.**
(DOCX)

**S1 Fig. Sequence alignment of all 73 CHS proteins of Solanum specie.** Color bars on the left represent the 10 groups in Fig 4. Active site residues are highlighted in yellow, malony-CoA binding sites are highlighted in blue and other conserved sequence are shown in green.
(JPG)

**S2 Fig. Interaction network key to anthocyanin biosynthesis in eggplant.** The pink labels represent the CHS gene family.
(PDF)

## Author Contributions

**Data curation:** Xiaohui Liu, Aidong Zhang.

**Resources:** Jing Shang, Zongwen Zhu, Dingshi Zha.

**Software:** Jing Shang.

**Supervision:** Xiaohui Liu, Jing Shang.

**Validation:** Zongwen Zhu, Dingshi Zha.

**Visualization:** Shengmei Zhang, Jing Shang, Zongwen Zhu.

**Writing – original draft:** Xuexia Wu.

**Writing – review & editing:** Xuexia Wu, Aidong Zhang, Dingshi Zha.

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
