## [Decision Letter · Decision Letter 0]

27 Jan 2020

PONE-D-19-33168

Genome-wide analysis of chalcone synthase (CHS) family from eggplant (Solanum melongena L.) in flavonoid biosynthetic pathway and expression pattern in response to heat stress

PLOS ONE

Dear Dr. Wu,

Thank you for submitting your manuscript to PLOS ONE. After careful consideration, we feel that it has merit but does not fully meet PLOS ONE’s publication criteria as it currently stands. Therefore, we invite you to submit a revised version of the manuscript that addresses the points raised during the review process.

We would appreciate receiving your revised manuscript by Mar 12 2020 11:59PM. To enhance the reproducibility of your results, we recommend that if applicable you deposit your laboratory protocols in protocols.io, where a protocol can be assigned its own identifier (DOI) such that it can be cited independently in the future. For instructions see: http://journals.plos.org/plosone/s/submission-guidelines#loc-laboratory-protocols

We look forward to receiving your revised manuscript.

Kind regards,

Muthamilarasan Mehanathan, Ph.D.

Academic Editor

PLOS ONE

Journal Requirements:

https://journals.plos.org/plosone/article?id=10.1371/journal.pone.0119054

https://link.springer.com/article/10.1007%2Fs11032-017-0721-x

In your revision ensure you cite all your sources (including your own works), and quote or rephrase any duplicated text outside the methods section. Further consideration is dependent on these concerns being addressed.

'This work was supported by the Agricultural Committee Basic Project (Shanghai Agricultural word (2015) No 6–2-3), the National Key Technology R&D Program during the 13th Five-Year Plan Period (2017YFD0101904) and the China Agriculture Research System (Grant No. CARS-25). The funding bodies did not play a role in the

design of the study and the collection, analysis, 413 and interpretation of data or in the composition of the manuscript.'

'YES-Specify the role(s) played'

Please provide an amended Funding Statement that declares *all* the funding or sources of support received during this specific study (whether external or internal to your organization) as detailed online in our guide for authors at http://journals.plos.org/plosone/s/submit-nowPlease state what role the funders took in the study.  If any authors received a salary from any of your funders, please state which authors and which funder. If the funders had no role, please state: "The funders had no role in study design, data collection and analysis, decision to publish, or preparation of the manuscript."

'I have read the journal's policy and the authors of this manuscript have the following competing interests'

a. Please complete your Competing Interests statement to state any Competing Interests.

If you have no competing interests, please state "The authors have declared that no competing interests exist.", as detailed online in our guide for authors at http://journals.plos.org/plosone/s/submit-now

6. Please amend either the title on the online submission form (via Edit Submission) or the title in the manuscript so that they are identical.

7. PLOS requires an ORCID iD for the corresponding author in Editorial Manager on papers submitted after December 6th, 2016. Please ensure that you have an ORCID iD and that it is validated in Editorial Manager. To do this, go to ‘Update my Information’ (in the upper left-hand corner of the main menu), and click on the Fetch/Validate link next to the ORCID field. This will take you to the ORCID site and allow you to create a new iD or authenticate a pre-existing iD in Editorial Manager. Please see the following video for instructions on linking an ORCID iD to your Editorial Manager account: https://www.youtube.com/watch?v=_xcclfuvtxQ

8. Please amend either the abstract on the online submission form (via Edit Submission) or the abstract in the manuscript so that they are identical.

Reviewers' comments:

Reviewer's Responses to Questions

**Comments to the Author**

1. Is the manuscript technically sound, and do the data support the conclusions?

Reviewer #1: Yes

Reviewer #2: No

2. Has the statistical analysis been performed appropriately and rigorously? 

Reviewer #1: No

Reviewer #2: No

3. Have the authors made all data underlying the findings in their manuscript fully available?

Reviewer #1: Yes

Reviewer #2: No

4. Is the manuscript presented in an intelligible fashion and written in standard English?

Reviewer #1: Yes

Reviewer #2: No

5. Review Comments to the Author

Reviewer #1: In this manuscript entitled ‘Genome-wide analysis of chalcone synthase (CHS) family from eggplant (Solanum melongena L.) in flavonoid biosynthetic pathway and expression pattern in response to heat stress’ the authors have identified CHS-encoding genes of eggplant, performed their in silico characterization and analysed the expression pattern of key genes associated with anthocyanin biosynthesis pathways. Here are the following comments on the research for the authors:

1. The manuscript need to be revised for its English, grammar and flow of information. For example in L130: ‘The profiles of CHS (PF00195 and PF02797) wwere downloaded from the Pfam’- wwere need to be corrected to were. Line 198: ‘the other 7 Solanaceae species ranged from 17.3 to 47.4, the length of protein ranged’ kDa need to be added to the molecular weights mentioned here.

2. It would be more informative for the readers if authors will provide a comparative mapping of CHS genes in the genome of other related plants where sequence information are available.

3. Fig. 7. represents key anthocyanin biosynthesis gene expression profiles in response to heat stress. A bar diagram with proper error bars for individual gene will be the most appropriate way for presentation of this result. The authors are advised to prepare the same for readers convenient.

4. Authors have performed the phylogenetic classification of CHS gene in Solanaceae and found that 73 CHS genes have classified into 10 groups. However, it has not been mentioned or discussed that on what basis the CHS are classified and how one group differs with the others.

5. As authors have mentioned in the text about some of the cis regulatory element present on all the SmCHS promoter, similarly they should mention the unique cis element for each SmCHS genes. Not all CHS genes had shown similar expression pattern, thus a probable correlation of cis regulatory element on promoter and expression pattern of the gene should be mentioned in the discussion.

Reviewer #2: The manuscript entitled “Genome-wide analysis of chalcone synthase (CHS) family from eggplant (Solanum melongena L.) in flavonoid biosynthetic pathway and expression pattern in response to heat stress” by Wu et al. reports the in silico analysis of chalcone synthase (CHS) gene family in eggplant. In brief, the current manuscript presents biochemical characteristics, phylogenetic relationships, gene structures, cis-regulatory elements, regulatory network and functional predictions of the CHS gene family members.

The current quality of the manuscript leaves a number of questions to be answered, which are outlined below. The exposition of the results and its discussion in the manuscript needs a major revision including careful English usage and style edition.

1. The manuscript title starts with ‘Genome-wide analysis’; however, most of the analysis has been performed on CHS proteins. The work performed in the manuscript does not justify the title.

2. The ‘Introduction’ section of the manuscript is not coherent. Also, it does not describe the relevance of studying CHS gene family in proper manner.

3. One of the major concerns is, authors have given RNA-Seq data in the result section. However, the source of this data is not described in the “Material and Method” section.

4. Also, authors have not provided the proper statistics of the validation experiment. Statistical analysis should be done to evaluate the significance of qRT-PCR data. It is recommended to include statistical analysis in the ‘Materials and methods’ section.

5. Writing in Subheadings 4 to 7 under the ‘Results’ section is very technical. The data is not described properly. Authors have chosen to explain the dataset in the ‘Discussion’ section. The outcome of the experiments should be described in the ‘Results’ section and interpretation in consistence with the previous studies should be given in the ‘Discussion’ section.

6. In Figure 3A, the abundance of cis-elements is given in all the CHS genes in eggplant. How the regulation of all the genes can be shown together?

7. In Figure 5, there is no downregulation value. How the data has been calculated? What control has been taken and how the data has been normalized?

8. Figure 6, network is not very clear. A clear image has to be provided.

9. In ‘Material and Method’ section, what is CK? Why the samples harvested at 28º C are termed as CK and all the subsequent data is compared to CK?

10. In Figure 7, all the genes are showing high expression at 3 h CK. Do authors have any explanation of these results? Also, this data has been recorded from which tissue?

11. The other major issue is that this manuscript requires a thorough language editing since there are numerous grammatical errors including dropped articles, split infinitives, improper word usage etc. It is advisable that the manuscript must be edited by an English-speaking personal.

12. Reference section has to be rechecked. For example, in few places journal name is abbreviated (J Exp Bot) and in few places it is not (Plant physiology). In few references, journal name is capitalized and in others it is not.

13. Biochemical analysis of CHS proteins (pI, molecular weight etc.) is not relevant to the manuscript. This data can be removed from the draft.

Overall, the manuscript is not of adequate quality. The manuscript should be revised thoroughly for data presentation, result interpretation, description and language.

6. PLOS authors have the option to publish the peer review history of their article (what does this mean?). If published, this will include your full peer review and any attached files.

Reviewer #1: Yes: ROSHAN KUMAR SINGH

Reviewer #2: No

---

## [Author Response · Author response to Decision Letter 0]

5 Mar 2020

Dear Reviewers

Thank you and anonymous reviewers very much for your kind and useful comments for our manuscript “Genome-wide analysis of chalcone synthase (CHS) family from eggplant (Solanum melongena L.) in flavonoid biosynthetic pathway and expression pattern in response to heat stress” (PONE-D-19-33168). We have read and seriously considered the comments very carefully, checked and revised the manuscript many times. Major revisions throughout the revised manuscript have been made as followings.

1. The title name have been changed to “Chalcone synthase (CHS) family analysis from eggplant (Solanum melongena L.) in flavonoid biosynthetic pathway and expression pattern in response to heat stress”

2. The ‘Introduction’ section has been modified according to the comments

3. The results of all 73 CHS proteins sequence alignment, conserved residues and sequence diversity were analyzed. 

4. We corrected some other grammatical errors and words spelling mistakes. 

The followings are responses to the comments point-by-point. 

Review Comments to the Author

Reviewer #1: In this manuscript entitled ‘Genome-wide analysis of chalcone synthase (CHS) family from eggplant (Solanum melongena L.) in flavonoid biosynthetic pathway and expression pattern in response to heat stress’ the authors have identified CHS-encoding genes of eggplant, performed their in silico characterization and analysed the expression pattern of key genes associated with anthocyanin biosynthesis pathways. Here are the following comments on the research for the authors:

1. The manuscript need to be revised for its English, grammar and flow of information. For example in L130: ‘The profiles of CHS (PF00195 and PF02797) were downloaded from the Pfam’- wwere need to be corrected to were. Line 198: ‘the other 7 Solanaceae species ranged from 17.3 to 47.4, the length of protein ranged’ kDa need to be added to the molecular weights mentioned here. 

Answer: Thank you for pointing out the mistakes in the manuscript. We have corrected these issues in the revised manuscript according to comments, and checked the manuscript many times. Please refer to line 163 in revised manuscript with track changes.

2. It would be more informative for the readers if authors will provide a comparative mapping of CHS genes in the genome of other related plants where sequence information are available.

Answer: Chromosomal location of CHS genes were added in Table S1. Sequence alignment of all 73 CHS proteins of Solanum specie showed in the Fig S1. The Ⅷ, Ⅸ and Ⅹ groups are distinguished from other groups mainly depends on the position 1-164 amino acids, GroupsⅠ, Ⅱ and Ⅲ are relatively conservative at the position 260-360 amino acids, in which the other groups are very diverse(S1 Fig). Please refer to line 325-328. 

3. Fig. 7. represents key anthocyanin biosynthesis gene expression profiles in response to heat stress. A bar diagram with proper error bars for individual gene will be the most appropriate way for presentation of this result. The authors are advised to prepare the same for readers convenient. 

Answer: Thanks for your suggestion. Error bars have been added in the Fig7.

4. Authors have performed the phylogenetic classification of CHS gene in Solanaceae and found that 73 CHS genes have classified into 10 groups. However, it has not been mentioned or discussed that on what basis the CHS are classified and how one group differs with the others. 

Answer: The results of all 73 CHS proteins sequence alignment were added in S1 Fig. Conserved residues and sequence diversity of 73 CHS proteins were analyzed. Please refer to line 240-244 and 325-328

5. As authors have mentioned in the text about some of the cis regulatory element present on all the SmCHS promoter, similarly they should mention the unique cis element for each SmCHS genes. Not all CHS genes had shown similar expression pattern, thus a probable correlation of cis regulatory element on promoter and expression pattern of the gene should be mentioned in the discussion. 

Answer: I'm sorry we miss the point. We have added this to the ‘result’ section refer to line 303-307.

Reviewer #2: The manuscript entitled “Genome-wide analysis of chalcone synthase (CHS) family from eggplant (Solanum melongena L.) in flavonoid biosynthetic pathway and expression pattern in response to heat stress” by Wu et al. reports the in silico analysis of chalcone synthase (CHS) gene family in eggplant. In brief, the current manuscript presents biochemical characteristics, phylogenetic relationships, gene structures, cis-regulatory elements, regulatory network and functional predictions of the CHS gene family members.

The current quality of the manuscript leaves a number of questions to be answered, which are outlined below. The exposition of the results and its discussion in the manuscript needs a major revision including careful English usage and style edition.

1. The manuscript title starts with ‘Genome-wide analysis’; however, most of the analysis has been performed on CHS proteins. The work performed in the manuscript does not justify the title. 

Answer: We changed the manuscript title as to ‘Chalcone synthase (CHS) family members analysis from eggplant (Solanum melongena L.) in the flavonoid biosynthetic pathway and expression patterns in response to heat stress’.

2. The ‘Introduction’ section of the manuscript is not coherent. Also, it does not describe the relevance of studying CHS gene family in proper manner. 

Answer: We have added more information about CHS involve in synthesis of flavonoid compounds during heat stress defense. Please refer to line 70-80.

3. One of the major concerns is, authors have given RNA-Seq data in the result section. However, the source of this data is not described in the “Material and Method” section. 

Answer: The overview of RNA-seq result from our lab was published (Zhang S et al, BMC Plant Biol. 2019 (1): 1-13). We therefore added a brief description of data source about RNA-seq in the“Material and Method” section. Please refer to line 196-198.

4. Also, authors have not provided the proper statistics of the validation experiment. Statistical analysis should be done to evaluate the significance of qRT-PCR data. It is recommended to include statistical analysis in the ‘Materials and methods’ section. 

Answer: we have added the statistical analysis information of qRT-PCR in the ‘Materials and methods’ section. Please refer to line 223-225.

5. Writing in Subheadings 4 to 7 under the ‘Results’ section is very technical. The data is not described properly. Authors have chosen to explain the dataset in the ‘Discussion’ section. The outcome of the experiments should be described in the ‘Results’ section and interpretation in consistence with the previous studies should be given in the ‘Discussion’ section. 

Answer: We agree and have revised the ‘results’ and ‘discussion’ accordingly. Please refer to line 351-361. 

6. In Figure 3A, the abundance of cis-elements is given in all the CHS genes in eggplant. How the regulation of all the genes can be shown together?

Answer: We changed the histogram of figure 3A into a table to help readers understand the results.

7. In Figure 5, there is no downregulation value. How the data has been calculated? What control has been taken and how the data has been normalized?

Answer: Some errors have been corrected in line 351 of manuscript. Three SmCHS genes (SmCHS1, SmCHS2, and SmCHS3) were continuously downregulated under 38 ℃ and 45 ℃ treatment compared with the CK (27 ℃). Gene expression level was estimated by RNA-seq from mean FPKM (fragments per kilobase of exon model per million reads mapped) values for each treatment, and showed the expression patterns in heatmap. Each treatment had three biological replicates.

8. Figure 6, network is not very clear. A clear image has to be provided.

Answer: Is the network image you referred to Figure S2? We're sorry that there are too many genes in the network image. The picture can be expanded up to 6400%, so that the above words can be read clearly.

9. In ‘Material and Method’ section, what is CK? Why the samples harvested at 28º C are termed as CK and all the subsequent data is compared to CK?

Answer: The optimum growth temperature for eggplant is between 22 and 30°C. Eggplants subjected to high temperature may exhibit to stagnation of growth, abortion of flower buds, and decrease of pollen viability rate and fruit set, and the peel’s color will turn light when the temperature is over 35°C. Therefore, samples of 27 °C were termed as CK.

10. In Figure 7, all the genes are showing high expression at 3 h CK. Do authors have any explanation of these results? Also, this data has been recorded from which tissue?

Answer: Under heat stress, SmCHS4 and some anthocyanin biosynthesis related genes show different expression profiles at 38 ℃-3h. These results suggest that these co-up-regulated genes contribute to protect the eggplant at beginning of heat stress defense. These data were recorded from the eggplant peels, which is indicated in the Figure 5 and Figure 6.

11. The other major issue is that this manuscript requires a thorough language editing since there are numerous grammatical errors including dropped articles, split infinitives, improper word usage etc. It is advisable that the manuscript must be edited by an English-speaking personal.

Answer: We checked and revised the manuscript many times. Some grammatical errors and words spelling mistakes have been modified by English editing company of American Journal Experts (AJE). 

12. Reference section has to be rechecked. For example, in few places journal name is abbreviated (J Exp Bot) and in few places it is not (Plant physiology). In few references, journal name is capitalized and in others it is not.

Answer: This issue has been corrected.

13. Biochemical analysis of CHS proteins (pI, molecular weight etc.) is not relevant to the manuscript. This data can be removed from the draft.

Overall, the manuscript is not of adequate quality. The manuscript should be revised thoroughly for data presentation, result interpretation, description and language.

Answer: Biochemical analysis of CHS proteins was removed.

---

## [Decision Letter · Decision Letter 1]

23 Mar 2020

PONE-D-19-33168R1

Chalcone synthase (CHS) family members analysis from eggplant (Solanum melongena L.) in the flavonoid biosynthetic pathway and expression patterns in response to heat stress

PLOS ONE

Dear Dr. Wu,

Thank you for submitting your manuscript to PLOS ONE. After careful consideration, we feel that it has merit but does not fully meet PLOS ONE’s publication criteria as it currently stands. Therefore, we invite you to submit a revised version of the manuscript that addresses the points raised during the review process.

We would appreciate receiving your revised manuscript by May 07 2020 11:59PM. To enhance the reproducibility of your results, we recommend that if applicable you deposit your laboratory protocols in protocols.io, where a protocol can be assigned its own identifier (DOI) such that it can be cited independently in the future. For instructions see: http://journals.plos.org/plosone/s/submission-guidelines#loc-laboratory-protocols

We look forward to receiving your revised manuscript.

Kind regards,

Muthamilarasan Mehanathan, Ph.D.

Academic Editor

PLOS ONE

Reviewers' comments:

Reviewer's Responses to Questions

**Comments to the Author**

1. If the authors have adequately addressed your comments raised in a previous round of review and you feel that this manuscript is now acceptable for publication, you may indicate that here to bypass the “Comments to the Author” section, enter your conflict of interest statement in the “Confidential to Editor” section, and submit your "Accept" recommendation.

Reviewer #1: All comments have been addressed

Reviewer #2: All comments have been addressed

2. Is the manuscript technically sound, and do the data support the conclusions?

Reviewer #1: Yes

Reviewer #2: Yes

3. Has the statistical analysis been performed appropriately and rigorously? 

Reviewer #1: Yes

Reviewer #2: No

4. Have the authors made all data underlying the findings in their manuscript fully available?

Reviewer #1: Yes

Reviewer #2: Yes

5. Is the manuscript presented in an intelligible fashion and written in standard English?

Reviewer #1: Yes

Reviewer #2: No

6. Review Comments to the Author

Reviewer #1: (No Response)

Reviewer #2: The manuscript entitled “Chalcone synthase (CHS) family member analysis from eggplant (Solanum melongena L.) in the flavonoid biosynthetic pathway and expression patterns in response to heat stress” by Wu et al. reports the in silico analysis of chalcone synthase (CHS) gene family followed by the expression analysis in eggplant.

First, I would appreciate the authors that they have addressed all the queries stated in the previous review. The quality of the manuscript has improved a lot. However, there are still few questions unanswered. Also, there are so many grammatical errors in the manuscript that altogether changes the meaning of the study. I have highlighted all the comments in the manuscript itself.

Please revise the manuscript once again for the grammatical corrections and other queries stated in the manuscript pdf file.

7. PLOS authors have the option to publish the peer review history of their article (what does this mean?). If published, this will include your full peer review and any attached files.

Reviewer #1: No

Reviewer #2: No

---

## [Author Response · Author response to Decision Letter 1]

31 Mar 2020

The followings are responses to the comments point-by-point.

1. line 20. “isprimarily”

Answer: a space was added between the two words.

2. line 30-31. “occurrence of the duplicated words, “Under heat stress”

Answer: we removed duplicate words.

3. line 32. Wrong presentation of “at 38 ℃3h”

Answer: we have corrected.

4. line 33. “CHS protein biochemical characteristics”, deleted content in the result.

Answer: We have used evolutionary relationship instead of biochemical characteristics.

5. line 36-37. “SmCHS showed two or more expression patterns and performed multiple functions to regulate anthocyanin content. Combined with analysis of regulatory networks, to the results of this study may facilitate further research to understand the regulatory mechanism governing peel color in eggplants.”. Grammatical errors.

Answer: We have revised the sentences. We are sorry about our errors.

6. line 42-43. “subject to high”. Grammatical error.

Answer: We have corrected according to comments.

7. line 44. “decrease of pollen”.

Answer: We have revised it.

8. line 47. “molecular mechanism governing high temperature stress”

Answer: We have rewritten the sentence.

9. line 66. “for instance”

Answer: We have deleted the two words.

10. line 68-69. “in bread wheat”.

Answer: We have revised the manuscript.

11. line 72. Full-width comma is behind of the word “oak”.

Answer: We have revised it.

12. line 74. “CHSV and CHSVII”

Answer: We have changed italics to non-italics.

13. line 101-102. 'fruit shape has a' is not sounding proper.

Answer: We have revised the manuscript.

14. line 103. This line is not meant for the materials. It can be written into the Introduction section.

Answer: We have deleted the sentence.

15. line 141. “boostarp test”, Grammatical error.

Answer: We have used “Bootstarp value” instead of “bootstrap test”.

16. line 186. 1 μg of what? Please mention.

Answer: We have revised the manuscript according to the suggestion above.

17. Line 208. “The the length”, “the” repeated.

Answer: We have deleted the duplicate word according to the comment.

18. Line 209-210. Which other solanacae species?

Answer: We have added some details according to the comment.

19. Line212-214. What exactly as done after taking an average of the amino acids? It is still not clear

Answer: According the comment, we have rewritten the sentence.

20. Line 226-227. Is there any literature available citing this functional diversity for SmCHS 7?

Answer: We have added two references according the comment.

21. Line 250. Please replace 'and' with 'whereas'

Answer: We have corrected the error.

22. Line 252. 'cytoplasm' instead of 'cytoplasmic'

Answer: We have corrected the error.

23. Line 255 Table 2. 'Width' or "Length', please check once. Also, please add the unit. For example, in this case, it must be 'amino acids'

Answer: We have used “Length” instead of “Width” and have added the unit.

24. Line 257. Please describe Figure 3A first. Please try to maintain the sequence throughout the manuscript.

Answer: We have adjusted the order of the two subgraph (Fig3A and Fig3B) in Fig 3. we also have revised the manuscript.

25. Line 266-277. Please add 'element' after 'elicitor response'. Binding site or what? Please specify.

Answer: We have corrected them according the comments.

26. Line 293. Please add space before 'is'

Answer: We have revised the manuscript according the comment.

27. Line 300. Why SmCHS4 is in italics, but not the others. Moreover, the author is talking about protein sequences in this line. Do we really need to italicize it?

Answer: We have revised it.

28. Line 304. 'Colored' instead of 'color'

Answer: We have corrected the error.

29. Line 309. This RNAseq data was obtained from which tissue. Please specify.

Answer: We have added details of tissue.

30. Line 312. Why authors have chosen to write the name of these to genes in the list of genes not identified in the study?

Answer: We want to show that the CHS gene families may have tissue-specific or functionally differentiation. We have added this inference to the end of this paragraph.

31. Line 313-314. In this line, it is written that few of the SmCHSs were not identified. However, in the heatmap figure, expression data of all the CHS gene is given. How is that possible?

Answer: In order to show all genes related to anthocyanin biosythesis, the expression level of undetected genes were set to 0. These undetected genes were also shown in the heat map. We have rewritten these sentences.

32. Line 319. “at 45 ℃”

Answer: We have revised the manuscript.

33. Line 345. Please rewrite this line. It is not making any proper sense. Please add few lines in this paragraph to describe the link between this paragraph and SmCHSs.

Answer: We have added some details in this paragraph to describe the link between this paragraph and SmCHSs. 

34. Line 350. The function can not be involved in anything. Please rewrite this line.

Answer: We have rewritten this sentence according the comment.

35. Line 398. Please capitalize 'biosynthesis'. You may write 'stress factors' instead of 'stressors'.

Answer: We have corrected the errors.

36. Line 405. Please cite the reference properly with the year.

Answer: We have corrected it.

37. Line 422. Please remove full stop.

Answer: We have removed the full stop.

38. Line 424. What is this reference? Please describe.

Answer: We have revised the manuscript.

---

## [Editor Report · Decision Letter 2]

2 Apr 2020

Chalcone synthase (CHS) family members analysis from eggplant (Solanum melongena L.) in the flavonoid biosynthetic pathway and expression patterns in response to heat stress

PONE-D-19-33168R2

Dear Dr. Wu,

We are pleased to inform you that your manuscript has been judged scientifically suitable for publication and will be formally accepted for publication once it complies with all outstanding technical requirements.

With kind regards,

Muthamilarasan Mehanathan, Ph.D.

Academic Editor

PLOS ONE

---

## [Editor Report · Acceptance letter]

6 Apr 2020

PONE-D-19-33168R2 

Chalcone synthase (CHS) family members analysis from eggplant (*Solanum melongena* L.) in the flavonoid biosynthetic pathway and expression patterns in response to heat stress 

Dear Dr. Wu:

I am pleased to inform you that your manuscript has been deemed suitable for publication in PLOS ONE. Congratulations! Your manuscript is now with our production department. 

With kind regards,

on behalf of

Dr. Muthamilarasan Mehanathan 

Academic Editor

PLOS ONE